# Butyltin Contamination in Fishing Port Sediments after the Ban of Tributyltin Antifouling Paint: A Case of Qianzhen Fishing Port in Taiwan

**Shu-Hui Lee** [1,†], **Yung-Sheng Chen** [2,†], **Chih-Feng Chen** [3], **Frank Paolo Jay B. Albarico** [3,4], **Yee Cheng Lim** [3], **Ming-Huang Wang** [3], **Chiu-Wen Chen** [1,*] and **Cheng-Di Dong** [1,*]

1. General Study Center, National Kaohsiung University of Science and Technology, Kaohsiung 81157, Taiwan; a0553@nkust.edu.tw
2. CSBC Corporation, No.3 Jhonggang Road, Siaogang District, Kaohsiung 81234, Taiwan; 104983@csbcnet.com.tw
3. Department of Marine Environmental Engineering, National Kaohsiung University of Science and Technology, Kaohsiung 81157, Taiwan; dong3762@nkust.edu.tw (C.-F.C.); albaricofrankpaolojay@gmail.com (F.P.J.B.A.); yeecheng@nkust.edu.tw (Y.C.L.); b88070554@nkust.edu.tw (M.-H.W.)
4. Fisheries and Marine Research Station, College of Fisheries and Allied Sciences, Northern Negros State College of Science and Technology, Sagay 6122, Philippines
* Correspondence: cwchen@nkust.edu.tw (C.-W.C.); cddong@nkust.edu.tw (C.-D.D.)
† These authors contributed equally to this work.

**Abstract:** This study investigated the concentrations of monobutyltin (MBT), dibutyltin (DBT) and tributyltin (TBT) in the sediments of the Qianzhen Fishing Port (Taiwan) in 2020. Further, the pollution status, composition, and potential ecotoxicity of BTs were evaluated. This case study provides a reference for the benefits of the ban of TBT-based antifouling paint to date. Results showed that the total butyltin ($\Sigma$BTs, sum of TBT, DBT, and MBT) concentrations measured in the sediments of the Qianzhen Fishing Port ranged between 14.2–807 ngSn·g$^{-1}$ dw, with an average of 356 ± 305 ngSn·g$^{-1}$ dw. TBT was the most dominant species, with an average concentration of 303 ± 287 ngSn·g$^{-1}$ dw. This average TBT concentration is about 4.3 times lower than in 2003, showing the progress of gradual degradation of TBT in the sediments. Still, the degradation is rather slow, with a half-life of about 8.09 years. An analysis of the effects of TBT on organisms in the sediments of the Qianzhen Fishing Port was carried out according to the TBT toxicity guidelines of the US Environmental Protection Agency and the assessment class criterion for imposex (ACCI) of the Oslo and Paris Commission (OSPAR). The results showed that TBT levels in 80% of the sediments may pose negative effects on sensitive gastropods, and half of the sediments may even have an impact on gastropod reproduction. These show that marine life is still affected and threatened by TBT compounds, despite the decline of TBT concentrations since the ban of TBT-containing antifouling paints on ships in 2008. Therefore, it is necessary to continue paying attention to the changes of TBT concentrations and their potential ecological risks in the marine environment, and to formulate TBT management plans and strategies to mitigate their impacts in marine ecosystems.

**Keywords:** butyltins; fishing port; sediments; tributyltin (TBT); organotins

## 1. Introduction

Butyltins (BTs) are environmental hormones or endocrine disruptors, which are toxic and extremely stable compounds. Monobutyltin (MBT) and dibutyltin (DBT) are mainly used as light and heat stabilizers in the process of PVC production [1]. Tributyltin (TBT), which has bioinhibitory properties, has been used in antifouling paints on ship hulls since the 1960s to prevent organisms from attaching on the hull and hindering navigation [2]. Due to the excellent anti-biofouling benefits of TBT, from 1970 to 1980, TBT was widely used in antifouling paints, which also caused TBT to be directly released from the paint into

water bodies [2]. Since BTs are hydrophobic substances, they easily adsorbed on suspended particles after entering the water, and then deposited in the sediments. However, since BTs are easily desorbed, resuspension of the sediment leads to enhanced BT concentrations in the water column [3]. In marine waters, BTs can be mainly decomposed through microbial degradation and UV photolysis; however, only biodegradation can occur in sediments [4]. The exchange capacity of BTs in seawater is higher because they are more susceptible to light or biological degradation than in sediments. Therefore, the half-life of TBT in natural water bodies (water column) is 6 days to several months, while it is 1–8.7 years in marine sediments [2]. Moreover, the half-life of TBT is even longer in anaerobic sediments, reaching more than 10 years [5]. In other words, once the water environment is polluted by BTs, organisms will face the effects of long-term exposure and accumulation.

It is well known that exposure to TBT causes changes in the endocrine systems of marine organisms, most notably imposex in female gastropods [6,7]. This syndrome has been reported in more than 260 gastropods worldwide [8]. In addition, studies of negative impacts on marine life including oyster deformity, obesity metabolic syndrome (syndrome) in fish, and immunotoxicity and mortality of juvenile mussels have been reported [6,9–11]. Due to the high toxicity of BTs to non-target species, the International Maritime Organization (IMO) adopted the International Convention on the Control of Harmful Anti-Fouling Systems on Ships in 2001, proposing a global ban on TBT, which was globally enforced in September 2008 [12]. Taiwan has also followed IMO recommendations, banning the use of TBT-based antifouling paints on ships less than 25 m in length since 2003, and banning it for all ships since 2008. This banning resulted in reduced BT levels in water bodies and minimized the probability of biological imposex [2,13,14], moderating the concerns of TBT pollution from ships. However, recent reports on BT levels and their associated bio-androgenic effects still exist in marine environments around the world [6,7,15–19], especially in fishing ports, commercial ports, shipyards and marinas [6,7,20].

Taiwan, being surrounded by seas, is very active in marine fisheries due to its geographical location and surrounding ocean currents, forming a fishery rich environment. There are about 222 fishing ports along the coast of Taiwan, with a total of about 21,772 fishing vessels and nearly 1100 ocean-going fishing vessels [21]. Kaohsiung City in Taiwan has more than 3400 fishing boats, with a fishing output of about 500,000 metric tons, accounting for half of Taiwan's fishery production [22]. There are 16 fishing ports along the coast of Kaohsiung City, including the Qianzhen Fishing Port, the largest ocean-going fishing port in Taiwan. This intensive and frequent vessel traffic may lead to impacts on the ecology of the fishing port area [23], including the release of TBT from antifouling paints on ships [7]. Studies on BTs have become a concern in Taiwan with its growing fishing industry. The Taiwan Environmental Protection Agency selected 17 fishing ports to investigate the TBT content of sediments (average concentrations were between 200–9600 ngSn·g$^{-1}$ dry weight (dw)) in 2003 [24]. One of these fishing ports was highly contaminated with TBT (41–205 ngSn·g$^{-1}$ dw), while the other 16 were severely contaminated (>205 ngSn·g$^{-1}$ dw) [25]. Among the 17 fishing ports, Qianzhen Fishing Port was the second highest with an average TBT concentration of 1300 ngSn·g$^{-1}$ dw in sediments. This high TBT concentration indicated that large amounts of TBT were released from ship hull antifouling paints, which eventually accumulated in the sediments. However, since then, there have been no reports of BTs related to the sediments in Taiwan fishing ports.

Hence, the objectives of this study were to collect sediments from the Qianzhen Fishing Port and analyze them for particle size distribution, organic carbon content, and butyltin (TBT, DBT and MBT) concentrations. Likewise, the pollution status and composition of BTs in sediments and the potential ecotoxicological effects of TBT were evaluated. The results of this case study provide evidence on the effectiveness of the ban of TBT-containing antifouling paints to date, especially on the levels and ecotoxicological effects of TBT in fishing port sediments.

## 2. Materials and Methods

### 2.1. Study Area and Sediment Sampling

Qianzhen Fishing Port (22°34′11″ N, 120°18′49″ E) has the highest fishing vessel tonnage and catches and is the largest ocean-going fishing port in Taiwan. The fishing port area is adjacent to Kaohsiung Port, the largest commercial harbor in Taiwan, and the entrance/exit of the fishing port is close to the main waterway of Kaohsiung Port. Qianzhen Fishing Port has a berth of 27.07 hectares and a wharf of 3189 m [22]. It mainly provides fishing boats and an area for unloading, maintenance, refueling and supply operations. According to statistics, there are about 400 ocean-going fishing boats (fishing boats over 200 tons) in Qianzhen Fishing Port, including longline, squid, and purse seine fishing vessels [22]. In this study, 10 sediment sampling stations were established in the Qianzhen Fishing Port area (Figure 1). Surface sediments (0–10 cm) were collected on 17 July 2020 using stainless steel dredgers. The collected sediment was placed in a brown glass bottle (pre-rinsed with n-hexane) and sealed with a screw cap containing a Teflon gasket. Samples were stored in an ice bucket containing crushed ice until transported back to the laboratory. In the laboratory, sediment samples were freeze-dried (72 h), crushed with a mortar and pestle and impurities were picked out, placed in brown glass bottles, and stored in a freezer at −20 °C until further analysis.

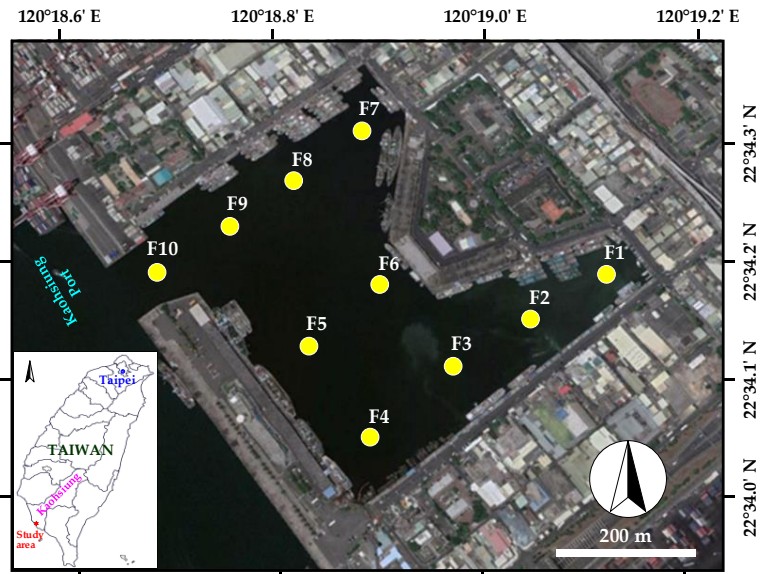

**Figure 1.** Map of sediment sampling sites in Qianzhen Fishing Port, Taiwan.

### 2.2. Sediment Sample Analysis

Sediment particle size analysis was performed with a laser particle size analyzer (Coulter LS230; Beckman Coulter, Inc., Brea, CA, USA). Total organic carbon (TOC) was analyzed using Walkley–Black titration method [26]. The pretreatment and analysis of BTs in sediments were based on the method suggested by Dong et al. [27]. A 5.0 g dried sediment sample was weighed and placed into a 40 mL brown glass bottle. Twenty (20) mL of acetate buffer (pH = 4.5), 0.2 mL of 10% sodium tetraethylborate (NaBEt$_4$), and 500 ng of the internal standard (Tetrapentyltin, TePeT) were then added. The solution was mixed using a vortex for 1 min, then added with 5 mL of 0.02% Tropolone/n-hexane, extracted with an ultrasonic device for 15 min, and centrifuged at 2000 rpm for 5 min. The upper organic phase was collected and placed in a 15 mL glass tube, then the extraction was repeated twice. The solution was mixed and added with copper wire to remove sulfur and water with anhydrous sodium sulfate, then blown with high-purity nitrogen at a slow speed to retain 0.5 mL. A 3 μL extract was collected for qualitative and quantitative analysis of BTs (TBT, DBT, and MBT) using gas chromatography-flame photometric detector (Agilent 4890D GC-FPD; Agilent Technologies, Santa Clara, CA, USA).

The quality control in this study included five-point calibration curve (0.05–1.0 ngSn/μL), process blank, replicate analysis, check standard and international reference standard PACS-2 (harbor sediment). The correlation coefficients (r) of the calibration curves were all greater than 0.995. Results of the program blank samples were all below the detection limits, and the relative difference percentage of the replicate analysis was less than 15%. The recovery of the check standards was between 93% and 105%. The method detection limits (MDL) for MBT, DBT and TBT were 0.5, 0.6 and 0.4 ng ngSn·g$^{-1}$ dw, respectively. The measured mean values ($n$ = 3) of TBT, DBT and MBT for the international reference standard PACS-2 (harbor sediment) were 878 ± 30, 1010 ± 30 and 502 ± 12 ngSn·g$^{-1}$ dw, respectively. The recoveries relative to certified values of TBT (890 ± 105 ngSn·g$^{-1}$ dw), DBT (1047 ± 64 ngSn·g$^{-1}$ dw) and MBT (600 ngSn·g$^{-1}$ dw, information value only) were 98.7%, 96.4% and 83.7%, respectively.

### 2.3. Data Analysis

Mud content, TOC and BT iso-concentration maps of sediments were drawn using Surfer(R) Version 10.0 (Golden Software, Golden, CO, USA). Considering the affinity of organic matter for organic pollutants, TOC was used to normalize BT concentrations in sediments [28]. Normalized BT (Nor. BTs) concentrations were calculated using the equation: Nor. BTs = BTs × 1%/TOC% [28]. If the measured value is lower than MDL, it is calculated as 1/2 MDL. The concentration of Nor. TBT in the sediments was compared with 2 screening values proposed by the US EPA, namely lower screening value (LSV; Nor. TBT = 5.15 ngSn·g$^{-1}$ dw) and higher screening value (HSV; Nor. TBT = 72.04 ngSn·g$^{-1}$ dw) [27,29]. In addition, the assessment of the ecotoxicological effects of TBT in sediments was carried out according to the assessment class criterion for imposex (ACCI) proposed at Oslo and Paris Commission (OSPAR) [30]. ACCI is converted from OSPAR's Coordinated Environmental Monitoring Program (CEMP) monitoring data. It is possible to integrate the assessment of imposex with concentrations of TBT in sediment [30]. ACCI is a 6 biological effect scale from A to F produced by integrating gastropod biological effects with sediment TBT concentrations as follows: Class A (not detected)—close to zero effects; Class B (<0.82 ngSn·g$^{-1}$ dw)—response caused by TBT concentrations below the ecotoxicological assessment criteria (EAC; 0.0041 ngSn·g$^{-1}$ dw), i.e., concentrations at which no adverse effects were observed in marine organisms; Class C (0.82–<20.5 ngSn·g$^{-1}$ dw)—level of response where females are not expected to be sterile; Class D (20.5–<82 ngSn·g$^{-1}$ dw)—sterile females are present in the population, but reproductively capable females remain; Class E (82–205 ngSn·g$^{-1}$ dw)—populations are unable to reproduce; and Class F (>205 ngSn·g$^{-1}$ dw)—populations of Nucella are absent/expired [27,30].

## 3. Results and Discussion

### 3.1. Concentrations of Butyltins in Surface Sediments

The average composition of mud (<63 μm) and sand (>63 μm) in Qianzhen Fishing Port sediments are 81.6 ± 10.3% and 18.4 ± 10.3%, respectively, indicating that the sediments are mainly composed of fine particles—with mud contents of 66.3–98.2% (Table 1). The fine particle deposition indicates that the Qianzhen Fishing Port is a low-energy water body, which is favorable for the adsorption and accumulation of pollutants. Higher TOC content was observed in sediment samples, averaging 7.3 ± 3.6% (Table 1). Fine-grained and rich organic properties may reduce oxygen availability in sediments, leading to low oxygen and/or anaerobic conditions [31]. The concentrations of MBT, DBT, TBT, ΣBTs and TBT contamination levels of the sediments are shown in Table 1. The concentration of MBT in the sediment was only 89.8 ngSn·g$^{-1}$ dw measured at the site F1, and the other sites were all below the detection limit of 0.5 ngSn·g$^{-1}$ dw. DBT was measured at sites F1–F2 and F4–F6, and its concentrations ranged from 5.4–291 ngSn·g$^{-1}$ dw, with an average of 67.0 ± 125.4 ngSn·g$^{-1}$ dw. Except for sites F7 and F10, TBT was detected at all sites, with concentrations ranging from 13.1–786 ngSn·g$^{-1}$ dw, and an average of

$303 \pm 287$ ngSn·g$^{-1}$ dw. The ΣBTs concentrations ranged from 13.1–807 ngSn·g$^{-1}$ dw, with an average of $356 \pm 305$ ngSn·g$^{-1}$ dw. According to the TBT-contaminated sediments class [25], sites F7 and F10 were non-contaminated (TBT < 1.2 ngSn·g$^{-1}$ dw), F9 was moderately contaminated (TBT = 8.2–41 ngSn·g$^{-1}$ dw), F1, F6 and F8 were highly contaminated (TBT = 41–205 ngSn·g$^{-1}$ dw), while other sites were severely contaminated (TBT > 205 ngSn·g$^{-1}$ dw).

**Table 1.** Distribution of particle size, TOC, and butyltins in surface sediments of Qianzhen Fishing Port.

| Item | F1 | F2 | F3 | F4 | F5 | F6 | F7 | F8 | F9 | F10 |
|---|---|---|---|---|---|---|---|---|---|---|
| Latitude (22° N) | 34′11.56″ | 34′09.18″ | 34′06.52″ | 34′02.94″ | 34′07.97″ | 34′10.83″ | 34′19.96″ | 34′16.91″ | 34′14.36″ | 34′11.72″ |
| Longitude (120° E) | 19′06.86″ | 19′02.28″ | 18′57.70″ | 18′53.37″ | 18′49.57″ | 18′54.00″ | 18′52.73″ | 18′49.12″ | 18′45.26″ | 18′40.78″ |
| Mud (<63 μm, %) | 66.3 | 73.3 | 87.9 | 86.6 | 88.9 | 89.0 | 73.1 | 82.7 | 98.2 | 70.2 |
| Sand (>63 μm, %) | 33.7 | 26.7 | 12.1 | 13.4 | 11.1 | 11.0 | 26.9 | 17.3 | 1.8 | 29.8 |
| TOC (%) | 11.8 | 13.9 | 4.38 | 8.66 | 6.76 | 5.7 | 8.42 | 7.48 | 3.89 | 2.11 |
| MBT (ngSn·g$^{-1}$ dw) | 89.8 | <0.5 [a] | <0.5 | <0.5 | <0.5 | <0.5 | <0.5 | <0.5 | <0.5 | <0.5 |
| DBT (ngSn·g$^{-1}$ dw) | 291 | 20.7 | <0.6 | 9.9 | 7.9 | 5.4 | <0.6 | <0.6 | <0.6 | <0.6 |
| TBT (ngSn·g$^{-1}$ dw) | 204 | 786 | 235 | 652 | 404 | 70.1 | <0.4 | 57.7 | 13.1 | <0.4 |
| ∑BTs (ngSn·g$^{-1}$ dw) [b] | 585 | 807 | 236 | 662 | 412 | 75.8 | – | 57.7 | 13.7 | – |
| TBT contamination class [c] | high | severe | severe | severe | severe | high | non | high | moderate | non |

[a] The measured value is below the detection limit, [b] ∑BTs: sum of MBT, DBT, and TBT. Measured values below the MDL are calculated as 1/2 the MDL, [c] TBT contamination class are assigned as non-contamination: <1.2; light: 1.2–8.2; moderate: 8.2–41; high: 41–205; severe: >205 ngSn·g$^{-1}$ dw [25].

The BTs (MBT, DBT, and TBT) in sediments of the Qianzhen Fishing Port were higher than most regions of the world (Table 2), but lower than those of the Port of Gdynia (Poland) [32–34], Port of Gdańsk (Poland) [31,33], and harbors and bays in Korea [35,36]. The degradation rate of TBT in sediments is slow, and even slower in anaerobic sediment conditions [1,37,38]. As shown by BT levels in sediments from different regions of the world (Table 2), semi-closed and relatively low-energy harbor or bay sediments have rather high BTs levels. This may be because they are usually anoxic or anaerobic and have low resuspension, which is not conducive to TBT degradation [38,39].

Several reports indicate that TBT concentrations in marine sediments have decreased after the ban of TBT-based antifouling paints, suggesting that the policy had a significant effect on reducing TBT concentrations in the marine environment [34,46,51]. The TBT concentration in the sediments in this study (303 ngSn·g$^{-1}$ dw on average) were approximately 4.3 times lower than that in 2003 (1300 ngSn·g$^{-1}$ dw on average) [24]. This result shows the progress of the gradual degradation of TBT in the sediments of the Qianzhen Fishing Port. This may be due to the degradation of TBT in the sediment by microbial activity and/or the release of TBT in the sediment into the water column and then photolysis by UV light [51]. Based on the two measured times (2003 and 2020) and the average concentration, the half-life of TBT in the sediments of the Qianzhen Fishing Port was calculated to be about 8.09 years. This is comparable to that in anaerobic sediments (about 10 years), which is longer than the half-life of TBT in aerobic sediments (1–2 years) [5]. It must be noted that despite the decrease in TBT concentrations in Qianzhen Fishing Port sediments, most are still classified as highly and severely contaminated (Table 1), which requires continuous monitoring [15,34,43,51–53].

### 3.2. Spatial Distribution and Composition of Butyltins

The iso-concentration maps of mud, TOC, TBT, Nor. TBT, ΣBTs and Nor. ΣBTs in the sediments of the Qianzhen Fishing Port are shown in Figure 2. The mud content of the sediments was higher in the middle of the port (83–98%), and lower at the entrance (F10) and inner parts of the port (F1, F2 and F7) (66–73%) (Figure 2a). The water depth of the wharf gradually increases from sites F1, F2, F7 to site F10, so the mud content of these sites was less, which may be affected by the seabed topography. Site F10 is located at the

entrance/exit of the port, which is narrow, so the flow velocity generated during high and low tides is relatively high, resulting to low mud content. The distribution of TOC was highest at sites F1 (11.8%) and F2 (13.9%), followed by sites F7 (8.42%) and F4 (8.66%), all of which are located at the corners of the port. The TOC concentrations decrease towards the port entrance (F10) (Figure 2b). The concentration distributions of TBT, Nor. TBT, ΣBTs and Nor. ΣBTs were higher in sites F2, F4 and F5, and decreased in the surrounding area (Figure 2c–f). Overall, TBT and BTs mainly accumulated in the sediments of sites F1–F5 and decreased towards sites F7–F10. The inner portion of the port (F1–F4) is the ship's refueling terminal, hence, leaks or accidental oil spills during refueling may result in increased organic loading in sediments. Sediments with high organic load may easily adsorb and accumulate organic pollutants such as BTs [54] in the area. In addition, high organic load should also be linked to anaerobic conditions, affecting BT degradations. This was corroborated by the significant positive correlation between sediment TOC and BTs ($r = 0.69$, $p < 0.05$, $n = 10$). The opposite side (F7–F10) of the port closer to the entrance/exit has better water exchange, resulting in lower accumulation of pollutants.

**Table 2.** Butyltin compounds in sediments (ngSn·g$^{-1}$ dw) from different regions of the world.

| Locations | Sampling Time | MBT | DBT | TBT | Reference |
|---|---|---|---|---|---|
| Port of Gdynia (Poland) | 2008 | 4–165 | 5–391 | 8–1910 | [32] |
| Port of Gdynia (Poland) | 2009 | 134–968 | 250–2716 | 1143–6408 | [33] |
| Port of Gdynia (Poland) | 2018 | 20.9–112 | 32.2–225 | 111–1120 | [34] |
| Port of Gdańsk (Poland) | 2008 | 7–684 | 9–2060 | 13–15780 | [32] |
| Port of Gdańsk (Poland) | 2018 | 40.4–490 | 80.1–889 | 122–1942 | [34] |
| Gulf of Gdańsk (Poland) | 2008 | 0.54–10.9 | 0.51–11.1 | 0.2–19.60 | [40] |
| Gulf of Gdańsk (Poland) | 2014 | 1.2–15.2 | 0.5–18.9 | 0.9–28.5 | [41] |
| Vistula Lagoon (Poland) | 2008 | 5.10–24.00 | <4.5–6.03 | 2.02–24.51 | [40] |
| Szczecin Lagoon (Poland) | 2008 | <1.0–33.97 | <1.2–38.66 | 2.39–97.98 | [40] |
| Oslofjord/Drammensfjord (Norway) | 2014 | 0.9–78.2 | 1.0–64.3 | n.d. [a]–145.0 | [41] |
| Nazaré Canyon (Portugal) | 2005 | <5.1–14 | <0.4–2.7 | <0.1–0.8 | [42] |
| Olhão (Portugal) | 2012 | 3.9 | 2.0 | 2.6 | [43] |
| Cagliari (Italy) | 2012 | 43.8 | 79.9 | 74.5 | [43] |
| El Kantaoui (Tunisi) | 2012 | 7.2 | 8.4 | 6.1 | [43] |
| Bahía Blanca Estuary (Argentina) | 2014 | <3.5–831 | <1.08–782 | <0.78–259 | [44] |
| Cape Town Harbor (South Africa) | 2011–2012 | n.a. [b] | n.a. | 10–829 | [45] |
| Todos os Santos Bay (Brazil) | 2010–2011 | <3–4.5 | 4–21 | <2–262 | [46] |
| Todos os Santos Bay (Brazil) | 2012 | <3–108 | <2–72 | <2–77 | [46] |
| Santos-São Vicente Estuarine (Brazil) | 2015 | <0.5–809 | <0.5–304 | <0.5–688 | [47] |
| Gulf of Mexico (Mexico) | 2018 | <0.3–195.8 | <0.3–79.0 | <0.3–90.6 | [7] |
| Sungai Pulai Estuary (Malaysia) | - | 6.5–12.2 | <0.1–6.1 | 8.1–10.6 | [48] |
| Masan, Haengam, and Gohyun Bay (Korea) | 2009 | 24–1048 | 22–455 | 13–1018 | [35] |
| Fishing port (Korea) | 2010 | 40–281 | 62–200 | 44–116 | [35] |
| Harbors (Korea) | 2010 | 15–6212 | n.d.–8747 | 3–55264 | [35] |
| Ulsan, Busan, and Gwangyang Bay (Korea) | 2014–2015 | <0.1–56.9 | <0.1–160 | <0.1–2304 | [36] |
| Fishing ports (China) | 2007 | <3.6–194 | <2.3–41.5 | <0.7–86 | [49] |
| Dumping sites along coastal (China) | 2012–2013 | n.d.–294.71 | 0.40–263.90 | 1.09–74.20 | [18] |
| South Hangzhou Bay (China) | 2013–2014 | n.d.–51.8 | n.d.–16.2 | n.d.–28.0 | [50] |
| Kaohsiung Harbor (Taiwan) | 2006 | n.d.–7.3 | n.d.–18.4 | 1.7–125 | [38] |
| Kaohsiung Harbor (Taiwan) | 2009 | 0.5–83.4 | 0.5–31.6 | 1.2–112 | [27] |
| Qianzhen Fishing Port (Taiwan) | 2003 | n.a. | n.a. | <2.4–7400 | [24] |
| Qianzhen Fishing Port (Taiwan) | 2020 | <0.5–89.8 | <0.6–291 | <0.4–786 | This study |

[a] n.d.: not detected. [b] n.a.: not available.

The distribution of MBT, DBT, and TBT in the Qianzhen Fishing Port normalized to 1% TOC in sediments is shown in Figure 3. Generally, the composition of Nor. BTs in the sediments was Nor. TBT dominated (74–98% of Nor. BTs). This, however, exclude sites F7 and F10 whose BTs were lower than the detection limit, and site F1 where Nor. DBT and Nor. MBT were the most dominant (accounting for 65% of Nor. BTs). In the

past, many studies have used the butyltin degradation index (BDI) to evaluate the fate and degradation of TBT in the environment, which is defined as BDI = (MBT + DBT)/TBT [55]. When BDI < 1, it indicates that the TBT of the sediment is a recent input, and conversely, the sediment TBT is an old input [55]. However, some studies have pointed out that the magnitude of the BDI value may represent a different situation. TBT persistence may be favored under certain conditions (e.g., hypoxia, darkness, low temperature, low microbial activity, and high levels of TBT), so, low BDI values, i.e., high proportions of TBT, appear to be the result of a slow degradation process [34,40]. DBT and MBT are mainly heat and light stabilizing additives used in the production of plastics. This may lead to the possible direct or indirect import of DBT and MBT into aquatic environments by rivers [27]. Therefore, a high BDI value with high proportion of MBT + DBT may also represent the input of DBT and MBT from other pollution sources [26,27,38]. Different BDI values may represent different sources of BTs or different degradation rates caused by different environmental conditions [27,56].

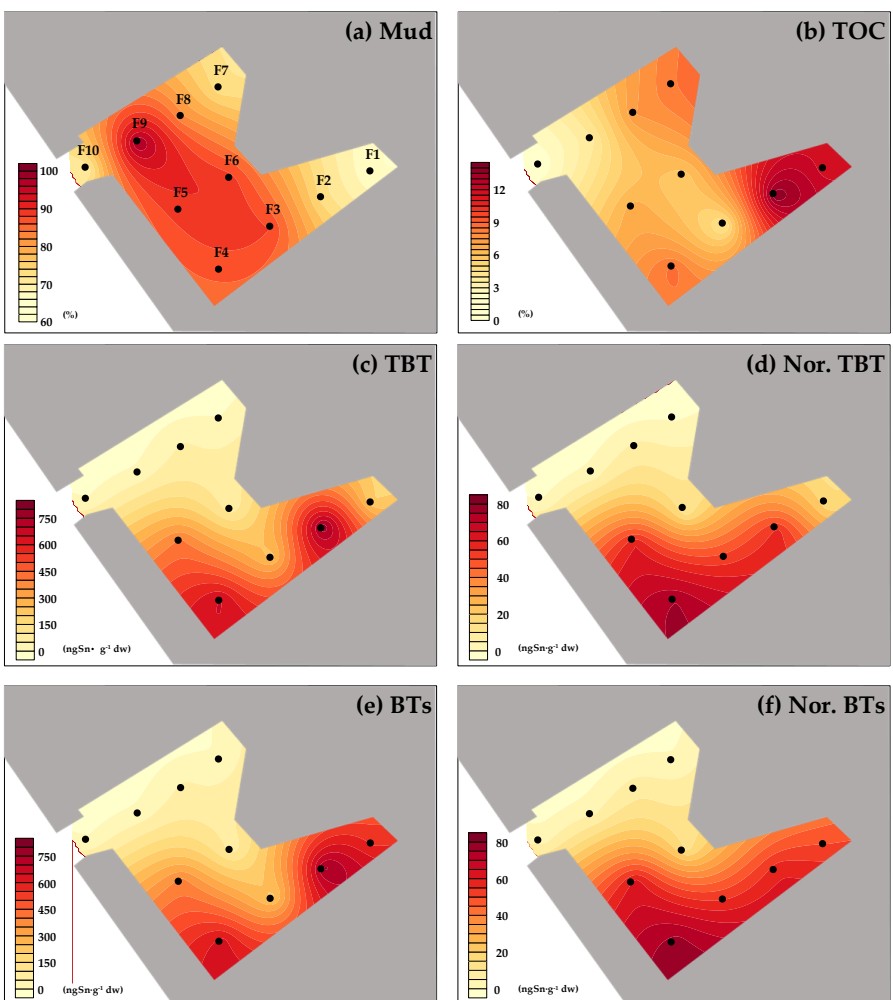

**Figure 2.** Iso-concentration plots of BTs in surface sediments of Qianzhen Fishing Port.

Figure 4 shows the MBT + DBT and TBT scatter plots in sediments of the Qianzhen Fishing Port. Except for site F1, the sediments exhibited high TBT and low MBT + DBT values with a significant linear relationship (slope = 0.021; r = 0.820, $p < 0.01$, $n = 9$). This indicates that the sources of BTs are similar and the degradation process of TBT in sediments is slow. However, it is not excluded that there are fresh TBT contamination inputs, such as TBT from ship paints. Site F1 has a relatively high MBT + DBT value and is far from the regression line (Figure 4), which means that the source of BTs at site F1 is different

from other sites, or there are inputs from other DBT and MBT sources other than from degradation of TBT. Site F1 is located in the deepest corner of the port area. It is usually in a low-flow state and becomes a gathering place for plastic waste. DBT and MBT additives may be released from these plastic wastes in the water body, and eventually accumulate in sediments.

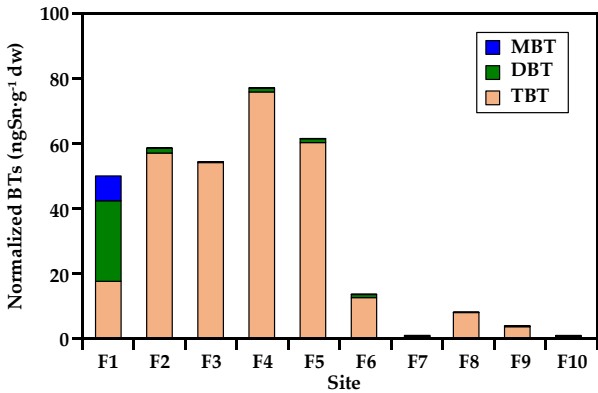

**Figure 3.** Composition of butyltins in the surface sediments of Qianzhen Fishing Port.

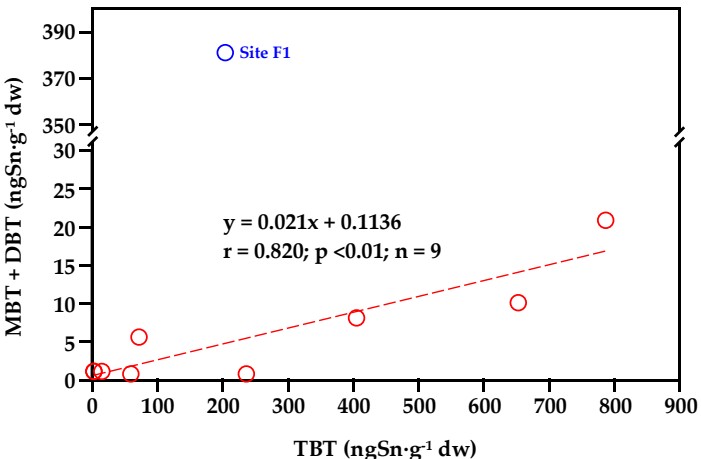

**Figure 4.** Relations between TBT and DBT + MBT in the surface sediments of Qianzhen Fishing Port.

### 3.3. Evaluation of the Contamination Levels and Ecological Toxicity of TBT

The concentration of Nor. TBT in the sediments of the Qianzhen Fishing Port were between <0.05–75.3 $ngSn·g^{-1}$ dw. The sediment Nor. TBT concentrations at sites F7 (<0.05 $ngSn·g^{-1}$ dw), F10 (<0.19 $ngSn·g^{-1}$ dw), and F9 (3.37 $ngSn·g^{-1}$ dw) were lower than LSV, indicating zero or negligible ecotoxicity. The Nor. TBT concentration (75.3 $ngSn·g^{-1}$ dw) in the sediments of site F4 were higher than the HSV value, indicating high ecotoxicity. The Nor. TBT concentrations of the other sites were between LSV and HSV, indicating the ecotoxicity is lower (Figure 5). In addition, according to the ACCI classification proposed by OSPAR [30], except for sites F7 and F10 which were classified as Class A (close to zero effects), the TBT concentrations of sediments at other sites were between Class C and F (Figure 6). Site F9, which was classified as Class C, may cause some negative biological effects due to higher TBT levels in the sediment than EAC. Sites F6 (Class D), F8 (Class D) and F1 (Class E) were classified as higher classes (D and E), indicating that sediment TBT levels may have reproductive effects in sensitive gastropods. Notably, sites F2–F5 were classified as the highest, Class F, indicating potential population threat in gastropods. In general, TBT levels higher than EAC were widely present in the sediments of the Qianzhen Fishing Port, and TBT levels in half of the sediments (sample) could affect the reproduction of

gastropods. This result suggests that TBT, from TBT-based antifouling paints, is persistent and not easily degraded in sediments [38].

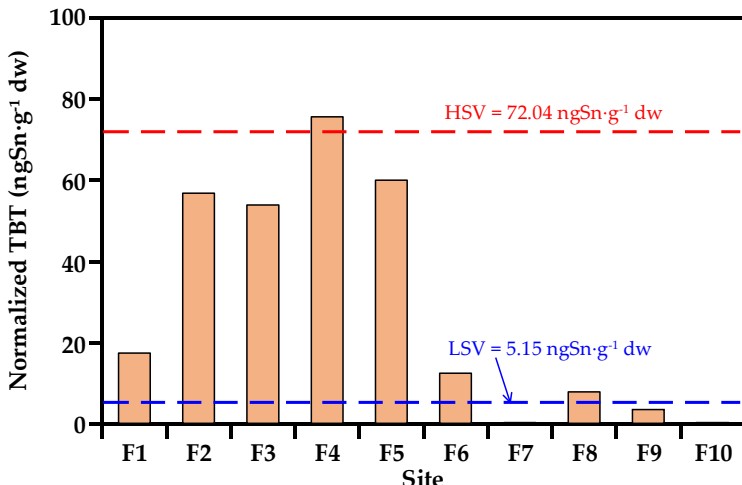

**Figure 5.** Distribution of Nor. TBT concentrations in the surface sediments of Qianzhen Fishing Port. Dashed line indicates the lower screening value (LSV) and higher screening value (HSV) level [29].

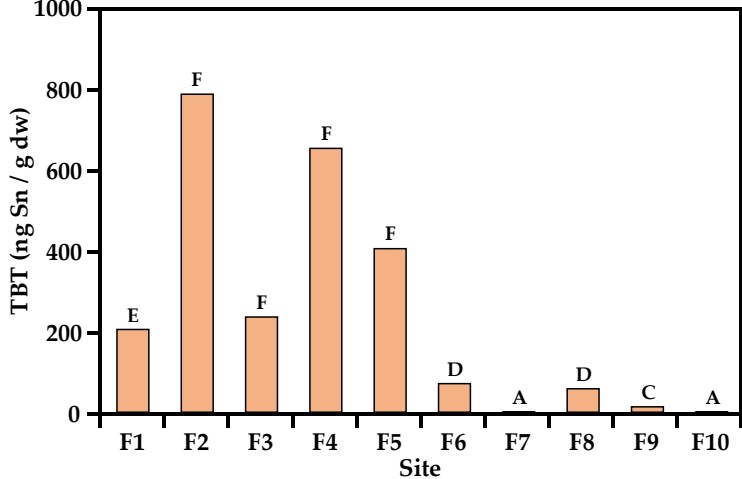

**Figure 6.** Assessment class criterion for imposex (ACCI) classification of TBT contained in Qianzhen Fishing Port sediments. English letters represent classes of ACCI classifies: Class A (not detected); Class B (<0.82 ngSn·g$^{-1}$ dw); Class C (0.82 to <20.5 ngSn·g$^{-1}$ dw); Class D (20.5 to <82 ngSn·g$^{-1}$ dw); Class E (82 to <205 ngSn·g$^{-1}$ dw); and Class F (>205 ngSn·g$^{-1}$ dw) [30].

It has been 12 years since IMO completely banned TBT antifouling paints, but the level of TBT in the sediments in this study and most other studies still pose reproductive hazards on marine organisms. Recent studies have found that the production of imposex in marine gastropods under the influence of TBT still exists [6,7]. Uc-Peraza et al. [7] also pointed out that there is still a high incidence of imposex in some gastropods near fishing ports and docks, which are closely associated with BT levels. Alarmingly, the TBT concentration in different marine sediments in the world is higher than Class C (0.82 to <20.5 ngSn·g$^{-1}$ dw), especially in the sediments of commercial ports, fishing ports and shipyards. Levels leading to serious threats against gastropod populations are >205 ngSn·g$^{-1}$ dw (Table 2). Many organisms in the ocean may be exposed to this endocrine disrupting compound and may be threatened with extinction. Therefore, TBT in the marine environment should be continuously monitored, evaluated, and properly managed.

The regular environmental dredging may increase the removal efficiency of TBT in sediments. This can lead to removal of accumulated pollutants in surface sediments, increasing

dissolved oxygen and resuspension, which are beneficial to increase microbial activities. Dredging is the simplest method to reduce TBT bioavailability and toxicity rapidly and significantly in aquatic systems. However, dredged sediment must be handled with care. In practice, it may be feasible to use dredged sediments for land reclamation or island building. On the one hand, new land can be obtained. Moreover, TBT biodegradation efficiency can be increased by enhancing microbial activity (such as injection well). In addition, it is necessary to educate fishermen against the use of TBT-containing boat paints, as it can induce fresh TBT contamination. An improved plan and strategic management for controlling TBT is further required. If the half-life of TBT in sediments is estimated to be 8.09 years according to this study, it will take at least 69 years (2089) for the TBT concentrations to drop to Class B (<0.82 ngSn·g$^{-1}$ dw), that is, when the biological effect caused by the TBT concentration is less than EAC.

### 4. Conclusions

This case study (butyltin contamination in sediments of Qianzhen Fishing Port after ban of TBT antifouling paints) clearly shows the progress of gradual degradation of TBT in marine sediments since the ban of TBT-containing antifouling paints in 2008. However, the degradation of TBT in sediments is slow (with an estimated half-life of 8.09 years), and current levels may still cause adverse effects on marine life. The results suggest that organisms in aquatic environments are still affected by TBT (endocrine disrupting substances) and may be threatened with species extinction. Therefore, it is recommended to carry out actions that are conducive for pollutant removal and increased microbial activities (such as regular environmental dredging), as well as to strengthen the ban of TBT anti-biological paint. Although the issue of TBT pollution is no longer a hot topic, current work and other recent research indicate that it is still a serious problem that requires immediate action by the authorities. Continuous monitoring, assessment, and development of pollution management strategies for TBT in the marine environment are necessary to ensure the sustainability of marine ecology.

**Author Contributions:** Conceptualization, S.-H.L., Y.-S.C. and C.-D.D.; methodology, C.-F.C.; validation, Y.C.L. and M.-H.W.; formal analysis, Y.C.L., M.-H.W. and F.P.J.B.A.; investigation, S.-H.L.; resources, C.-W.C. and C.-D.D.; data curation, Y.-S.C.; writing—original draft preparation, S.-H.L., Y.-S.C. and C.-F.C.; writing—review and editing, C.-F.C., C.-W.C. and F.P.J.B.A.; visualization, S.-H.L. and Y.-S.C.; supervision, C.-D.D.; project administration, C.-W.C.; funding acquisition, C.-D.D. All authors have read and agreed to the published version of the manuscript.

**Funding:** This research received no external funding.

**Institutional Review Board Statement:** Not applicable.

**Informed Consent Statement:** Not applicable.

**Data Availability Statement:** Not applicable.

**Conflicts of Interest:** The authors declare no conflict of interest.

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
