# Peer review of "Butyltin Contamination in Fishing Port Sediments after the Ban of Tributyltin Antifouling Paint: A Case of Qianzhen Fishing Port in Taiwan"

_water, doi:10.3390/w14050813_

Round 1
Reviewer 1 Report
- Proper Language correction must be required to improve it
- Reference are required in line 46-48.
- Add Latitude and longitude of the Qianzhen Fishing Port boundary
- State the objective and outcome of the present research at the end of the introduction section
- Add lat. and long. for all 7 sample locations
- Need to discuss about the effect on TBT on discussion part
- Highlight the typical source of TBT in study area
- Fig.6 need to explain well like why? reason? source? etc..
- Conclusion need to modify to satisfy the international readers
Reviewer 2 Report
The manuscript “Butyltins Contamination in Fishing Port Sediments after Ban of Tributyltin Antifouling Paint: A Case of Qianzhen Fishing Port in Taiwan” presents a clear research focused the contents of butyltins in sediments collected from a harbour that revealed high contents of these toxic compounds. It shows to what extent high contents of these compounds persist in sediments even after the application of norms to prohibit the application of paints that contain them. The number of samples is not sound, but the results obtained for distinct sediment features seem reliable. The research is well structured and in general easy to read.
As far as I see it, the manuscript looks more as a well-prepared report performed to evaluate an environmental issue than a real scientific paper. Anyway, we are used to see published similar works in scientific journals. Besides this point, I just have a few minor comments.
L. 53: Is “better” the adequate term? Maybe “higher”?
L. 170: Indicate the limits of clay, silt and sand. Note that when using a laser diffraction there is an overestimation of the size of tabular-shapes particles. Some authors estimate that a 2 micron clay particle will be measured by laser diffraction as if it was ~5.5 micron (e.g., Konert & Vandenberghe, 1997, Sedimentology: 44)
L. 233-234: Explain Nor meaning. Normalized to 1% TOC?
L. 238-239: High organic load should also be linked to anaerobic conditions, affecting BTs degradations
L. 297: Font change
L. 341-342: Rephrase
Finally, I found some repetitions that should be avoided. Namely, the information in lines 191-193, 206-207 and 310-312 were already presented in the information.
In summary, I recommend minor revision before acceptance in Water.
Reviewer 3 Report
Comments in the attachment

Round 2
Reviewer 1 Report
Author carried out a detailed correction and properly responded my quires